# Innovative Modeling Techniques and 3D Printing in Patients with Left Ventricular Assist Devices: A Bridge from Bench to Clinical Practice

**DOI:** 10.3390/jcm8050635

**Published:** 2019-05-09

**Authors:** Rishi Thaker, Raquel Araujo-Gutierrez, Hernan G. Marcos-Abdala, Tanushree Agrawal, Nadia Fida, Mahwash Kassi

**Affiliations:** 1Touro College of Osteopathic Medicine, Middletown, New York, NY 10940, USA; rkthaker@gmail.com; 2Houston Methodist DeBakey Heart & Vascular Center, Houston Methodist Hospital, Houston, TX 77030, USA; araujo@houstonmethodist.org (R.A.-G.); hgmarcosabdala@houstonmethodist.org (H.G.M.-A.); nfida@houstonmethodist.org (N.F.); 3Department of Internal Medicine, Houston Methodist Hospital, Houston, TX 77030, USA; tagrawal@houstonmethodist.org

**Keywords:** LVAD, computational fluid dynamic modeling, particle image velocimetry, 3D printing

## Abstract

Left ventricular assist devices (LVAD) cause altered flow dynamics that may result in complications such as stroke, pump thrombosis, bleeding, or aortic regurgitation. Understanding altered flow dynamics is important in order to develop more efficient and durable pump configurations. In patients with LVAD, hemodynamic assessment is limited to imaging techniques such as echocardiography which precludes detailed assessment of fluid dynamics. In this review article, we present some innovative modeling techniques that are often used in device development or for research purposes, but have not been utilized clinically. Computational fluid dynamic (CFD) modeling is based on computer simulations and particle image velocimetry (PIV) employs ex vivo models that helps study fluid characteristics such as pressure, shear stress, and velocity. Both techniques may help elaborate our understanding of complications that occur with LVAD and could be potentially used in the future to troubleshoot LVAD-related alarms. These techniques coupled with 3D printing may also allow for patient-specific device implants, lowering the risk of complications increasing device durability.

## 1. Introduction

Left ventricular assist device (LVAD) provides a bridge to transplant or destination therapy for patients with end-stage heart failure or low cardiac output. LVAD technology has evolved over the past 30 years. The first-generation models introduced a pulsatile flow mechanism, such as the Berlin Heart EXCOR and the Thoratec XVE, but suffered from high thrombosis and valve failure rates [1]. The second-generation LVADs, like the HeartMate II (Thoratec, Pleasanton, CA, USA), used axial impellers to deliver flow augmentation, but thrombotic and bleeding complications appeared to become more common in this iteration [1,2]. Third-generation centrifugal flow LVAD, the HeartWare HVAD (HeartWare Inc., Framingham, MA, USA) are smaller, frictionless, pulsatile, allow pericardial placement, and have fewer complications of in-pump thrombosis [3]. The latest device, HeartMate 3, demonstrated superiority in mortality, disabling stroke, or reoperation to replace or remove a malfunctioning device compared to second-generation LVADs in the MOMENTUM trial [3].

Yet, complications result from LVAD placement despite advances in pump design. Several complications, including GI bleed [4,5], stroke [6], pump thrombosis [7], and aortic regurgitation are thought to be secondary to altered fluid dynamics from LVAD. While echocardiography allows for useful hemodynamic assessment and cardiac computed tomography (CT) produces accurate anatomic visualization, flow characteristics and the dynamics of fluid motion are not well elaborated by these methods.

Several studies have been done with novel modeling techniques that allow assessment of flow characteristics that are unique to LVAD [8,9,10,11,12,13]. A vast majority of these studies are done in ex vivo models without patient specific modeling. We suspect that there is a gap between bioengineers and physicists developing modeling techniques and the clinicians who take care of patients. This review article aims to introduce clinicians to these novel simulation techniques, which predict LVAD flow. These novel techniques would facilitate troubleshooting for LVADs and also allow for preprocedure planning. The role of 3D printing is also highlighted, particularly in the realms of surgical implant techniques. Figure 1 summarizes the imaging techniques currently available and the modeling that could potentially be used for LVAD. 

## 2. Computational Fluid Dynamics

Computational fluid dynamics (CFD), to put quite simply, is a technique that utilizes the concepts of applied physics and computer simulations to understand “how fluid flows”. The specialized software simulates fluid motion within a virtual flow laboratory. CFD is considered to be a blood flow imaging technique that calculates flow based on certain assumptions rather than flow measurement [14,15]. CFD has been increasingly utilized in the realm of cardiovascular diseases. Most recently, the utilization of CFD in performing fractional flow reserve from coronary computed tomography (CT) data for evaluation of coronary artery disease was found to be incremental to coronary CT data alone. This technique is now being used clinically in evaluation of significant stenosis [16].

CFD is used in designing and validating LVAD prototypes. Prior to the Momentum 3 trial, the HeartMate 3 device was tested against HM II and showed low shear stress and better hemocompatibility profile [10]. However, it has not developed a role yet in day-to-day clinical practice. This may be due to the elaborate workflow algorithm and assumptions involved, which are two important limitations of this technique [17].

Several steps are involved in the workflow algorithm. Figure 2 summarizes the workflow for CFD in a patient with LVAD. Prior to running a computer simulation, the accurate 3D geometry of the object needs to be defined, usually from cardiac CT or 3D echocardiography. The physical characteristics of the fluid, in this case blood, and the surrounding boundaries are defined. In these computer simulations, blood is assumed to act as an ideal fluid. The quantity of blood flow through the “inlet” (the left ventricle) and the “outlet” (the aorta) can be derived from cardiac catheterization or echocardiography. These geometric and boundary conditions are inputs to a computer simulation that essentially functions as a virtual flow lab. The information can then be used to generate information on shear stress, heat dissipation, and wall pressure. Models for hemolysis and platelet activation can also be generated to test a device for hemocompatibility [10]. Figure 3 is an example of CFD simulations for varying outflow cannula angles. 

Fluid–structure interaction (FSI) is the study of fluid flow with a surrounding deformable structure. Several LVAD CFD studies have evaluated this interaction of flow from outflow cannula with the aorta. The CFD simulations allow quantitative assessment of hemodynamic parameters, such as pressure, velocity, wall shear stress (WSS), and displacement. An important observation from these studies is that the outflow cannula in the descending aorta generates highly disturbed flow in the ascending aorta compared to the location of outflow graft in the ascending aorta. This disturbed flow is thought to have a direct relation with atherosclerosis and thrombogenesis. Outflow graft implant in the descending aorta has fallen out of favor in clinical practice and is rarely performed. There are however, no large clinical studies looking at various ascending aorta graft locations with altered flow patterns and direct patient outcomes particularly stroke, which remains a source of immense morbidity and mortality in patients with LVAD [6]. Hence, CFD can be utilized to determine ideal inflow and outflow cannula position that would prevent adverse events, particularly stroke and development of aortic regurgitation [13,18,19,20]. CFD allows for evaluation of flow characteristics and feasibility of innovative graft positions, such as positioning of outflow graft in the subclavian artery [21]. Hemocompatibility has remained a big area of interest. One recent study looked at the degradation of von Willebrand factor (VWF) in continuous flow LVAD and determined that turbulence remained an important factor in VWF degradation regardless of exposure time with the device [22]. 

Most of the studies done with CFD, however, lack patient specific geometry and long-term outcomes. CFD can prove to be a useful technique that could potentially be used clinically for LVAD implant planning, surveillance and troubleshooting of complications of patient with LVAD. Assessment of adverse fluid characteristics in relation to adverse long-term clinical outcomes may provide important insight into surgical implant techniques.

## 3. Particle Image Velocimetry

Particle image velocimetry (PIV) uses rapid sequential imaging to compute velocity vectors and elaborate fluid dynamics. The technique is useful in understanding the dynamic motions of fluids since laminar and turbulent velocities are measured in real-time. While CFD is a computer simulation, PIV modeling utilizes ex vivo models and both techniques have good agreement with each other [11]. A transparent “phantom” organ, laser, camera, “seeding particles”, and image processing software are required for set up (Figure 4) [23]. The transparent “phantom” is usually printed from silicon or another material with excellent optical clarity [24]. Seeding particles are reflective, microscopic (between 10 and 100 nm), and the same density as the fluid. The laser reflects against the seeding particles, and the high-resolution camera captures sequential images that can be used to impute velocity vectors.

There are two competing constraints in the fluid selection for a PIV study: the density of the fluid being modeled and the refractive index of the fluid in the experiment [24]. Relative advances in blood-mimicking fluid (BMF) technologies has produced fluids that reproduce the dynamic viscosity of blood and produce a fluid with refractive index to optically clear silicone, but this BMF does not have the exact density of blood. Of these competing constraints, matching the refractive index is the most important because fluid that is not refractive index-matched may produce optical distortions that prevent appropriate velocity computations (Figure 5).

PIV has shown promise in understanding the unique fluid dynamics of LVAD placement. PIV was instrumental in in vitro modeling the pulsatile Lavare cycle while the HeartWare device was being developed, which lead to a reduction in the stagnation index and pump stasis, as well as a reduction in strokes and right ventricular failure clinically [26]. PIV has also been used to model wall shear stress that may be implicated in LVAD-associated acquired von Willebrand’s disease [27]. A PIV study of vortex formation and stasis elaborated on altered fluid dynamics in pre- and post-LVAD implantation [28].

When fluid dynamics of normal cardiovascular function is altered by the introduction of inflow and outflow cannulas, in vivo analysis is difficult since cannula positions are not standardized. By studying the velocity vectors associated with various cannula positions, we can impute the effects on hemodynamics and shear forces, and model troubleshooting options for clinical practice.

## 4. 3D Printing

3D printing is increasingly being used to model a variety of cardiovascular pathologies and is primarily used in surgical planning for congenital heart disease (CHD), transcatheter aortic valve replacement (TAVR), and tumor resections [29,30,31,32]. Several groups have created 3D models to aid in the placement of ventricular assist devices (VADs) utilizing imaging datasets from MRI angiography and cardiac CT [33]. Similarly, others have reported fitting and virtual implantation of total artificial hearts and VADs in children with heart failure [34,35,36]. Although these models have been useful for better anatomical positioning and surgical techniques, they lack the physiologic and hemodynamic simulation of continuous flow pumps and their impact on LVAD-specific complications. Therefore, there is an increased need to develop flow dynamic simulation models, with CFD or PIV modeling, in combination with patient-specific anatomic features.

The use of preprocedural 3D modeling replicating pathological flow conditions have been recently used in different settings. For example, 3D modeling for TAVR planning quantified the severity of valve insufficiency under controlled flow conditions [37]. Coarctation of the aorta has been modeled in 3D to allow for preprocedural troubleshooting [38]. Flow loops have also been created to assess the reproducibility of hemodynamics in 3D-printed models of patients with aortic stenosis utilizing a mock ventricle with compliance and resistance elements, pressure and flow transducers; these models correlate with in vivo measurements using pressure catheters and Doppler velocity profiles [29]. Russ et al. reported the use of 3D vascular phantoms connected to flow loops that simulated realistic circulatory conditions in endovascular intervention [39]. As 3D printing technology evolves, there is hope to develop patient-specific models that replicate the anatomic and physiological features to be used in LVAD surgical planning. Figure 6 is an example of a mock circulatory flow loop with compliance chambers and resistance chamber. 

## 5. Discussion

The paradigm of end-stage heart failure has evolved with the advent of left ventricular assist devices. Although, current LVAD technology is known to prolong survival, it is still associated with significant adverse events [40].These adverse events are thought to be a result of altered fluid dynamics that may in part be a result of surgical implant techniques. To mitigate the risk of altered fluid flow, it is imperative to have a fair understanding of fluid dynamics. CFD and PIV modeling allow accurate determination of fluid alterations. 

Creating an ex vivo 3D LVAD model in combination with CFD and PIV technologies would generate patient-targeted simulation of surgical techniques, elucidate optimal inflow and outflow cannula positioning, and provide exploration and preprocedural physician training of functional mock flow loops under patient-specific hemodynamic constraints. This form of modeling could establish optimal end-stage heart failure treatment in a personalized manner and limit long-term complications of LVAD.

However, it must be emphasized that prior to being used in clinical practice, it is extremely important to validate the findings from these modeling techniques. For instance, the findings from CFD can be validated against 3D models with mock circulatory flow loops. Prior to CT-fractional flow reserve (FFR) being used clinically, several studies were performed to validate against 3D models and coronary FFR [41]. Another important consideration is to consolidate the workflow of these techniques, as these models could be time-consuming. Simulations with these models are based on several assumptions and require input of data such as LVAD flow or timing of aortic valve opening, particularly in case of LVADs. Therefore, the accuracy of data derived from these techniques is heavily dependent on accurate input and assumptions that are as close as possible to actual human physiology and anatomy. 

Currently, these modeling techniques are underutilized in clinical practice as there is a gap between applied mathematicians, physicists, and cardiologists. Our aim in this review article was to simplify these techniques for clinicians taking care of LVAD patients to enhance collaboration between the various disciplines. Our intention is not to oversimplify the techniques and it must be emphasized that accurate performance requires expertise from specialists in each field. However, enhancing knowledge about the utility of these techniques can open doors for the various disciplines to work together. 

In the long run, improved fluid dynamics may help improve implant techniques and lessen the burden of significant adverse events. 

## Figures and Tables

**Figure 1 jcm-08-00635-f001:**
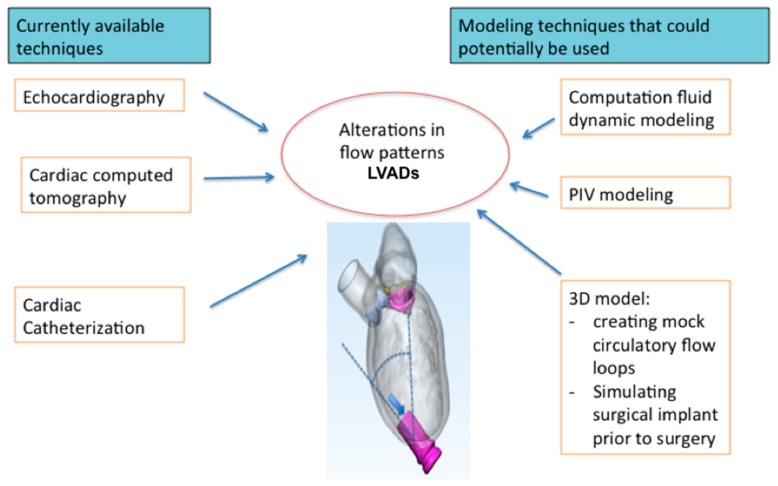
An elaboration of the currently available imaging modalities and modeling techniques that could be potentially employed in the future study of left ventricular assist device (LVAD).

**Figure 2 jcm-08-00635-f002:**
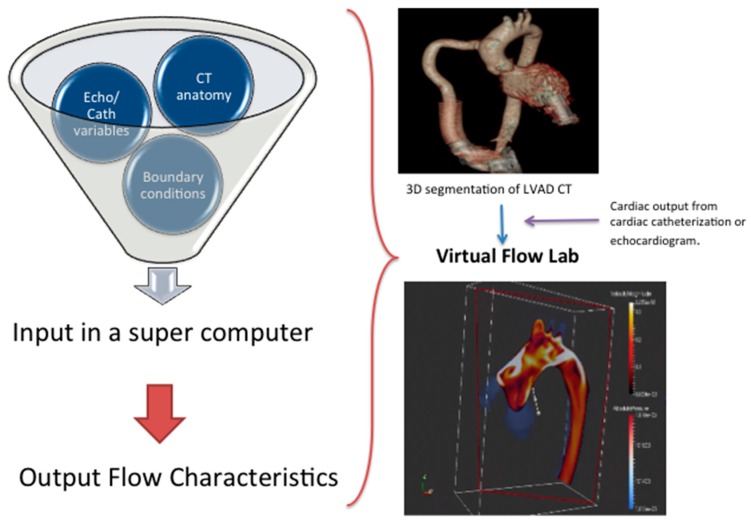
The workflow for computational fluid dynamics (CFD) analysis in a patient with LVAD.

**Figure 3 jcm-08-00635-f003:**
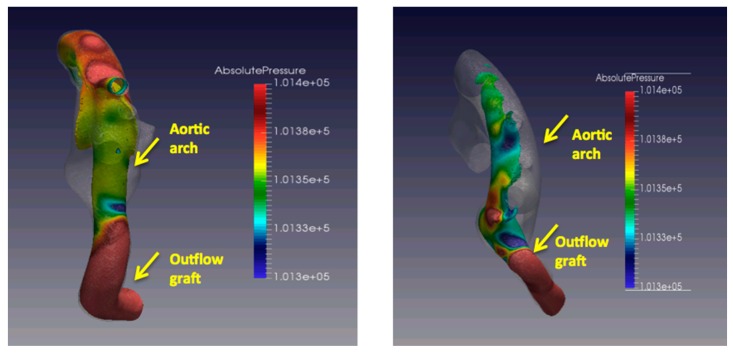
CFD simulations with varying outflow cannula angles relative to the aortic arch and variations in absolute pressure (in Pascals) and distribution on the aortic arch and great vessels.

**Figure 4 jcm-08-00635-f004:**
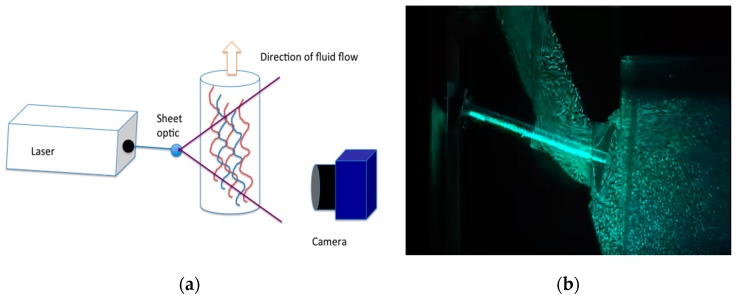
(**a**) Particle image velocimetry (PIV) requires a camera, laser, seeding particles, and optically clear “phantom.” The laser refracts off of the seeding particles and sequential images elucidate specific fluid dynamics; (**b**) Real-time snapshot of a PIV model. Seeding particles are illuminated by the laser and a high resolution camera takes sequential images. Data is analyzed by PIV software. (Courtesy of University of Brighton Youtube Channel [23]).

**Figure 5 jcm-08-00635-f005:**
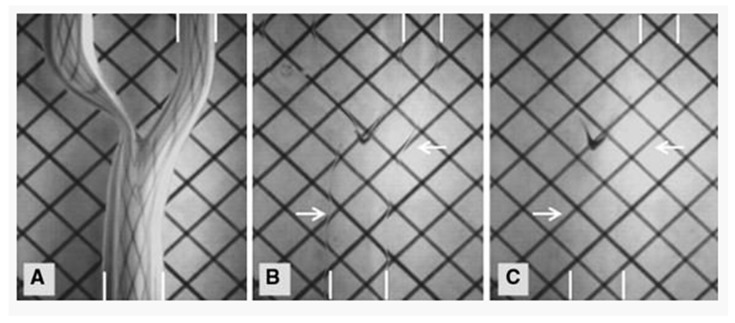
(**A**) The phantom is empty and has the refractive index of air. (**B**) The phantom contains “nearly-matched” fluid with minor optical distortions still present. (**C**) The phantom contains optimally matched fluid. (Courtesy Yousif et al. [25]).

**Figure 6 jcm-08-00635-f006:**
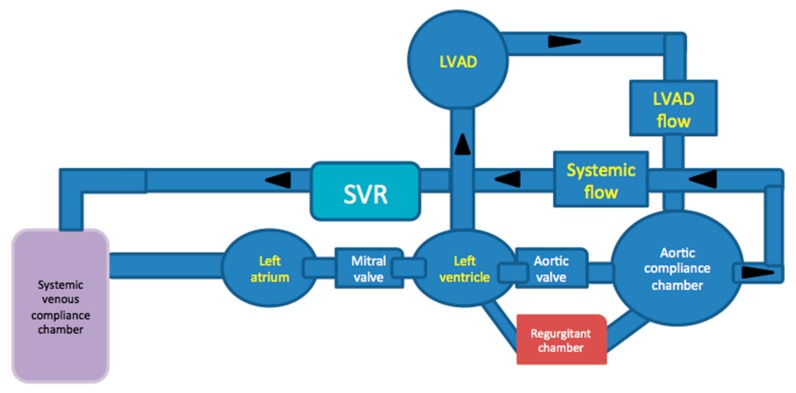
An example of a flow circulatory flow loop with LVAD connected to compliance chambers and flow meters.

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
