# Peer review of "Innovative Modeling Techniques and 3D Printing in Patients with Left Ventricular Assist Devices: A Bridge from Bench to Clinical Practice"

_jcm, 2019, doi:10.3390/jcm8050635_

Reviewer 1 Report

Timely review: concise and straight to the point with an appropriate selection of references.

Fluent and easy to read.

I have spotted these areas in need of attention:

Page 2, line 75: "practise" should be replaced with "practice"

Remove "degradation" from line 100, page 4: it has been repeated twice

Page 6, line 169: it is Fig. 6 and not Fig. 5

Page 6, line 188: again, "practise" should be replaced with "practice"

Author Response

We appreciate the reviewer's kind comments. 

All the errors mentioned have been corrected with track changes.

Reviewer 2 Report

 This review present some innovative modeling techniques that are often used in device development or for research purposes: Computational fluid dynamic (CFD) modeling and Particle image velocimetry. Overall, the paper is well designed but some aspects of the paper must be improved.

In figure 1 other thechnique can be added, i.e. X-ray angiographies. See Gaudio L.T., De Rosa, S., Indolfi C., M.V. Caruso, G. Fragomeni “HEMODYNAMICALLY NON-SIGNIFICANT CORONARY ARTERY STENOSIS: A PREDICTIVE MODEL” Proceedings of the International Workshop on Innovative Simulation for Health Care, 18-20 September 2017 Barcellona (Spain) ISbn: 978-88-97999-89-8 

The resolution of Figure 3 si too low and the description si too poor (i.e. the variable description and the numerical scale are missing).

Row 95, the authors can add the following reference in order to introduce also FSI (Fluid Structure Interaction) simulations: R. Mazzitelli, F. Boyle, E. Murphy, A. Renzulli, G. Fragomeni “Numerical prediction of the effect of aortic Left-Ventricular-Assist-Device outflow-graft anastomosis location”  Biocybernetics and Biomedical Engineering 36 (2016) 327-343 DOI:10.1016/j.bbe.2016.01.005 

The CFD simulations are also helpfull to estimate WSS parameter (i.e. OSI, RRT, TAWSS, and more) to draw clinical conclusions. See the following papers:

R. Mazzitelli, F. Boyle, E. Murphy, A. Renzulli, G. Fragomeni “Numerical prediction of the effect of aortic Left-Ventricular-Assist-Device outflow-graft anastomosis location”  Biocybernetics and Biomedical Engineering 36 (2016) 327-343 DOI:10.1016/j.bbe.2016.01.005

LEE B.K., Computational fluid dynamics in cardiovascular disease, Korean Circulation Journal, 2011, 41(8), 423–430, DOI: 10.4070/kcj.2011.41.8.423. 

M.V. Caruso, R. Serra, P. Perri, G. Buffone, F.G. Caliò, S. de Franciscis,  G. Fragomeni “A computational evaluation of sedentary lifestyle effects on carotid hemodynamics and atherosclerotic events incidence” Acta of Bioengineering and Biomechanics  Vol. 19 (3) DOI 10.5277/ABB-00682-2016-03

Gaudio L.T., Veltri P., De Rosa S., Indolfi C., Fragomeni G. “Model and Application to support the Coronary Artery Diseases (CAD): development and testing” Interdisciplinary Sciences: Computational Life Sciences - 2019 - DOI: w

Row 161, the authors can explain the acronym TAVR.

Row 162, 3D modeling and printig have ben used also in aorta coartaction surgery, see: L.T. Gaudio, P. Veltri, G. Fragomeni “Modeling and application of aorta coarctation: support system for pre-operative decision” BIBM 2018 IEEE International Conference on Bioinformatics and Biomedicine 2-6 december 2018 Madrid (Spain)

The discussion section is too poor. The authors can explain the importance of results validation in order to reach good results helpfull for clinical and surgical applications.

Author Response

Point 1: This review present some innovative modeling techniques that are often used in device development or for research purposes: Computational fluid dynamic (CFD) modeling and Particle image velocimetry. Overall, the paper is well designed but some aspects of the paper must be improved.

Point 1: We appreciate the reviewer’s kind comments. Please review the point by point changes that were made, including changes made to the discussion.

Point 2: In figure 1 other thechnique can be added, i.e. X-ray angiographies. See Gaudio L.T., De Rosa, S., Indolfi C., M.V. Caruso, G. Fragomeni “HEMODYNAMICALLY NON-SIGNIFICANT CORONARY ARTERY STENOSIS: A PREDICTIVE MODEL” Proceedings of the International Workshop on Innovative Simulation for Health Care, 18-20 September 2017 Barcellona (Spain) ISbn: 978-88-97999-89-8

Point 2: Thank you for sending us this reference. The reference is useful but is not relevant to our review as we are solely focusing on patients with Left ventricular assist devices. The reference is a useful modeling technique in coronary stenosis. If we add this to the image, we are afraid that it may mislead the clinicians.

Point 3: The resolution of Figure 3 si too low and the description si too poor (i.e. the variable description and the numerical scale are missing).

Point 3: We have elaborated the description for Figure 3.  In a separate file, we are sending the higher resolution of the image.

Point 4: Row 95, the authors can add the following reference in order to introduce also FSI (Fluid Structure Interaction) simulations: R. Mazzitelli, F. Boyle, E. Murphy, A. Renzulli, G. Fragomeni “Numerical prediction of the effect of aortic Left-Ventricular-Assist-Device outflow-graft anastomosis location”  Biocybernetics and Biomedical Engineering 36 (2016) 327-343 DOI:10.1016/j.bbe.2016.01.005

The CFD simulations are also helpfull to estimate WSS parameter (i.e. OSI, RRT, TAWSS, and more) to draw clinical conclusions.

R. Mazzitelli, F. Boyle, E. Murphy, A. Renzulli, G. Fragomeni “Numerical prediction of the effect of aortic Left-Ventricular-Assist-Device outflow-graft anastomosis location”  Biocybernetics and Biomedical Engineering 36 (2016) 327-343 DOI:10.1016/j.bbe.2016.01.005

LEE B.K., Computational fluid dynamics in cardiovascular disease, Korean Circulation Journal, 2011, 41(8), 423–430, DOI: 10.4070/kcj.2011.41.8.423.

M.V. Caruso, R. Serra, P. Perri, G. Buffone, F.G. Caliò, S. de Franciscis,  G. Fragomeni “A computational evaluation of sedentary lifestyle effects on carotid hemodynamics and atherosclerotic events incidence” Acta of Bioengineering and Biomechanics  Vol. 19 (3) DOI 10.5277/ABB-00682-2016-03

Gaudio L.T., Veltri P., De Rosa S., Indolfi C., Fragomeni G. “Model and Application to support the Coronary Artery Diseases (CAD): development and testing” Interdisciplinary Sciences: Computational Life Sciences - 2019 - doi: 10.1007/s12539-018-0311-6.

Point 4: We have elaborated on some of the more technical aspects of CFD and quoted some of the aforementioned studies. However, we have maintained a simplistic approach as the idea of this review is to target the clinicians who need to get a broader understanding of utility of these modeling techniques and not be lost in the technical terms.

Point 5: Row 161, the authors can explain the acronym TAVR.

Point 5: This was already done in the first paragraph.

Point 6: Row 162, 3D modeling and printing have been used also in aorta coartaction surgery, see: L.T. Gaudio, P. Veltri, G. Fragomeni “Modeling and application of aorta coarctation: support system for pre-operative decision” BIBM 2018 IEEE International Conference on Bioinformatics and Biomedicine 2-6 december 2018 Madrid (Spain)

Point 6: This has been inserted.  

Point 7: The discussion section is too poor. The authors can explain the importance of results validation in order to reach good results helpfull for clinical and surgical applications.

Point 7: We have elaborated the discussion and have talked about the importance of validating results and elaborated the limitations of these techniques.

Reviewer 3 Report

It is a well written review article, although the clinical significance is still to be determined. For eg in CFD- the amount of data that will be required to accurately feed into various models might be so extensive that it might not be feasible and if corners are cut it will be a case of garbage in garbage out. 

I do not believe that these technologies are simplistic in nature and to that effect the statement of the authors (in the discussion section)that they have simplified it so that clinicians can find it useful in clinical practice is inappropriate and an overextension of what the goal of this paper appears to be. These technologies are not readily available in everyday practice of medicine and the training required for medical personnel to use and interpret the information acquired from such technologies will be considerable.

Nonetheless this is a good review reminding us of what is on the horizon and might find its place in mainstay medicine in the future.

Author Response

Reviewer 3

Comments and Suggestions for Authors

It is a well written review article, although the clinical significance is still to be determined. For eg in CFD- the amount of data that will be required to accurately feed into various models might be so extensive that it might not be feasible and if corners are cut it will be a case of garbage in garbage out.

I do not believe that these technologies are simplistic in nature and to that effect the statement of the authors (in the discussion section)that they have simplified it so that clinicians can find it useful in clinical practice is inappropriate and an overextension of what the goal of this paper appears to be. These technologies are not readily available in everyday practice of medicine and the training required for medical personnel to use and interpret the information acquired from such technologies will be considerable.

Nonetheless this is a good review reminding us of what is on the horizon and might find its place in mainstay medicine in the future.

We appreciate the reviewer’s comments and we agree that it is important to not oversimplify the techniques. We have elaborated the discussion and have talked about the importance of validating results and elaborated the limitations of these techniques.

Round  2

Reviewer 2 Report

The paper improved the paper following the suggestion and the paper is ready for publication. Only a minor change is necessary: In figure 3 the pressure unit must be inserted.